# Changes in Medication Prescribing Due to COVID-19 in Dental Practice in Croatia—National Study

**DOI:** 10.3390/antibiotics12010111

**Published:** 2023-01-07

**Authors:** Ivana Šutej, Dragan Lepur, Krešimir Bašić, Luka Šimunović, Kristina Peroš

**Affiliations:** 1Department of Pharmacology, School of Dental Medicine, University of Zagreb, 10000 Zagreb, Croatia; 2Department of Infectious Diseases, School of Dental Medicine, University of Zagreb, 10000 Zagreb, Croatia; 3University Hospital for Infectious Diseases “Dr. Fran Mihaljević”, 10000 Zagreb, Croatia; 4Department of Orthodontics, School of Dental Medicine, University of Zagreb, 10000 Zagreb, Croatia

**Keywords:** prescriptions, antibiotics, medications, analgesics, dentistry, coronavirus, SARS-CoV-2

## Abstract

The 2019 coronavirus pandemic (COVID-19) has affected clinical practice and, consequently, drug prescribing in dental practice. We investigated how the pandemic affected the prescribing behavior of dentists in Croatia. Data on prescribing practices for this study were provided by the Croatian Health Insurance Institute. The analysis included the number of prescriptions, costs, and the number of packages prescribed. The World Health Organization’s defined daily dose per 1000 inhabitants (DID) per day was used as an objective utilization comparison. During the first pandemic year, prescribing practice changed the most. Wide-spectrum antibiotics, analgesics, and antiseptics showed the highest trend in change. A statistically significant change in prescribing practices during the pandemic period was noted for amoxicillin with clavulanic acid, ibuprofen, and ketoprofen which showed an increase in trend, while cephalexin and diclofenac showed a statistically significant decrease. The highest increase in trend was recorded for azithromycin, at +39.3%. The COVID-19 pandemic has been associated with a marked increase in medication utilization, especially in the first year of the pandemic. The increase in wide-spectrum antibiotic classes needs to be addressed and regulated so that patients accept that antibiotics are not a substitute for dental treatment and dentists always start treatment with narrow-spectrum antibiotics regardless of specific times, as is the case with the pandemic.

## 1. Introduction

The COVID-19 pandemic caused significant disruption in all aspects of personal and professional life. In dentistry, after the World Health Organization declared the pandemic [1], Croatia suspended all non-urgent dental activities for two months [2]. According to the recommendations of the Croatian Association of Dental Medicine, urgent treatment included uncontrolled bleeding, severe toothache, cellulitis, the spreading of bacterial soft-tissue infection with intraoral or extraoral swelling, and trauma involving the facial bones potentially compromising the patient’s airway [3]. This suspension was necessary due to the specific nature of the dental profession, including the frequent performance of aerosol-generating procedures, as well as the proximity to the patient’s face [4,5]. After Croatia successfully responded to the first wave of the pandemic, and successfully suppressed transmission by the end of April, dental services were gradually normalized. In May 2020, the Chamber of Dental Medicine and the Association issued “Guidelines and protocols for the opening and work of dental offices and dental laboratories of the Croatian chamber of dental medicine” [6]. Additional guidelines such as “Handling of healthcare professionals in case of suspicion of COVID-19, a disease caused by a new coronavirus (SARS-CoV-2)” which intended to further regulate healthcare in general, including dental medicine, were regularly updated with new information [7]. The guidelines recommended the use of rubber dams, a high-volume evacuator, and mouthwash prior to dental care, as well as four-handed work, and mechanical barriers. In aerosol-generating procedures, the use of an N95 respirator (or similar device) was recommended, in addition to a face shield, disposable apron/gown, a cap, gloves, and physical separation measures to maintain social distance. To date, the pandemic has resulted in significant changes in dentists’ activities and patients’ behavior. Therefore, the aim of this study was to investigate the impact of the COVID-19 pandemic on prescribing trends in dental medicine in Croatia, in order to support the further planning and adaptation of dental services, both through specific as well as regular times.

## 2. Materials and Methods

Data on dental prescriptions and medications dispensed were obtained from the Croatian Health Insurance Fund (CHIF), a central government medical insurance agency. The basis for this analysis was all dental prescriptions funded by CHIF. According to the official drug utilization reports of the Agency for Medical Products and Medical Devices, this is the majority of prescriptions in Croatia (89.5%) [8], which is a realistic utilization rate. In this study, data on the number of prescriptions for the years 2019 to 2021 were analyzed, which included prescriptions for all major classes of medications prescribed by Croatian dentists, namely, antibiotics, opioid analgesics, non-opioid analgesics, antifungals, and antiseptics. The data included the number of prescriptions, the cost of medicines in national currency (Croatian kuna; HRK), and the number of packages prescribed. The data did not include private prescriptions or medicines dispensed to in-hospital patients. Analysis of annual prescriptions was performed using Microsoft^®^ Excel software. Descriptive statistics were used to present the results as means and percentages. For the quantitative analysis, a methodology based on the defined daily dose (DDD)/1000 inhabitants/day (DID) and Anatomical Therapeutic Chemical classification (ATC), according to the World Health Organization (WHO), was used [9]. The defined daily dose per 1000 inhabitants per day (DID) formula was used to calculate a standardized measure of medicine use at the national level. 

The mid-year population estimates were obtained from the Croatian Bureau of Statistics, the official national data provider [10]. The number of licensed dentists, and the number of patients admitted for dental or oral health reasons for each year considered, were obtained from the Croatian Institute of Public Health (CIPH) [11].

The data did not include personal information about the patients or the dentists. Nevertheless, the study was independently reviewed by the Ethical Board of the School of Dental Medicine, University of Zagreb, and approved under approval No. 05-PA-30-XI-11/2019.

The collected data were statistically analyzed using the statistical package for social sciences (SPSS) version 29.0 (Armonk, NY, USA: IBM Corp.). Descriptive statistics are presented numerically in tables, or graphically in figures. Due to the central limit theorem, a paired *t*-test was used for comparisons among time periods to test the differences between specific means (prescribed drugs in time intervals). Intervals of 2 years, from 2014/2015 to 2020/2021, were compared. Each interval was compared with the next to show the change in prescribing trend in successive intervals. At the end, the first interval was compared with the last to show the change during the whole observation period.

## 3. Results

### 3.1. Prescribing practice

During the analyzed period, the average annual number of all prescriptions issued by dentists was 459,837 and both the number of prescriptions and the total cost of prescriptions increased by 4% and 2.3%, respectively, compared with the previous year (Table 1). On average, a single dentist issued 179 prescriptions per year, for an average of 28 prescriptions per 100 insured patients who used some type of dental service. Utilization of the ten most commonly prescribed dental medications for 2014–2021 by DID is summarized in Table 2. The most commonly prescribed dental medications by the number of prescriptions issued for 2014–2021 are summarized in Table 3.

#### 3.1.1. Antibiotics

Antibacterial agents (J01 and P01) were by far the most frequently prescribed drugs, accounting for an average of 80% of all dental prescriptions. The first choice, as the most commonly prescribed antibiotic, was amoxicillin with clavulanic acid (co-amoxiclav), which accounted for an average of 64% of all prescriptions and 49% of all prescriptions (Table 2 and Table 3). The total number of antibiotics prescribed by dentists increased by 7.744 (2.4%) in the first pandemic year and 6.222 (1.7%) compared with 2019. The major increase was recorded for azithromycin in the first (39%) and second (9%) pandemic years but the change was not statistically significant (*p* = 0.055). A statistically significant increase in utilization during the pandemic period was observed for co-amoxiclav (*p* = 0.002) for both pandemic years and clindamycin for the first pandemic year (*p* = 0.043). For cephalexin, a statistically significant decrease was observed during the pandemic period (*p* = 0.044) (Figure 1).

#### 3.1.2. Pain relief medications

The second most prescribed class of medications was pain relievers, with ibuprofen being the first choice and an 11% increase in prescriptions compared with the previous year (Table 2 and Table 3). In contrast, naproxen had by far the greatest relative increase (43%), followed by opioid analgesics (15%). In the second pandemic year, ibuprofen and naproxen continued with higher utilization, while the other representatives of this medication class slowed and their utilization decreased. A statistically significant increase in utilization during the pandemic period was observed for ibuprofen (*p* = 0.013) and ketoprofen (*p* = 0.01), while diclofenac showed a statistically significant decrease (*p* = 0.023) (Figure 1). 

#### 3.1.3. Antiseptics

As expected, the use of antiseptics increased in the beginning, especially for antiseptics recommended in the COVID-19 guidelines for dental practice. These primarily included povidone iodine and hydrogen peroxide, which increased by 24.8% and 21%, respectively, in the first year of the pandemic, whereas the trend reversed in the second year. The results were not statistically significant.

#### 3.1.4. Antifungals

The last group of medications commonly prescribed in dentistry is antifungals. This is the only group where the utilization of each medication decreased throughout the observation period and results were not statistically significant.

## 4. Discussion

In this study, the impact of the COVID-19 pandemic on dental prescriptions in Croatia was investigated. We found that the number of prescriptions issued by dentists increased significantly in the 2020 pandemic year, and was less for 2021. Prescribing practices during the period under observation showed an increase in the total number of dental prescriptions by 4% in the first year and 2.3% in the second pandemic year.

As the COVID-19 pandemic initially led to an almost complete cessation of normal activities, after a series of studies and adjustments, each country issued its own guidelines to allow for the resumption of regular activities, both private and professional [5,13]. In dentistry, after the initial two months of complete cessation of all non-urgent procedures, practice was adjusted to the new guidelines and rules [6], bringing changes in behavior for both the dentists and the patients. In this study, further adjustment was observed for dental prescribing practices. 

Our data show that the antibiotics were the most commonly prescribed type of medications accounting for approximately 80% of all prescriptions, and increasing by 2.4% in 2020 and 1.7% in 2021. The same trend of an overall increase in antibiotic prescribing has been observed worldwide [14,15,16,17], mostly due to the delay of dental treatments. This jump might be explained by exceptional circumstances, such as a combination of restrictive measures and general avoidance of aerosol-generating procedures, where analgesics and antibiotics took the place of conservative treatment for a limited period of time, because patients did not have access to providers as discussed above. In such cases where no operative treatment was undertaken, dentists seemed to tend to prescribe a broader-spectrum antibiotic, which has been shown in several studies [18,19,20]. The most frequently prescribed antibiotic in our study was co-amoxiclav, while the trend of amoxicillin prescriptions in Croatia was decreasing (8%). At this point, it is important to emphasize that the antibiotic of first choice in dentistry according to the guidelines and recommendations for the treatment of odontogenic infections is amoxicillin alone [21,22,23,24]. Amoxiclav accounted for 48.3% of all prescriptions, and an average of 67% of all antibiotics prescribed in the pandemic period, with a statistically significant increase during the pandemic period. Surprisingly, the biggest increase in use among the antibiotics was recorded for azithromycin (39%, 9%). The use of azithromycin during the COVID-19 pandemic could be explained by a large number of interstitial pneumonias (the hallmark of moderate and severe COVID-19). In addition, early guidelines for the treatment of COVID suggested the use of a combination of hydroxychloroquine and azithromycin, presuming that antiviral and immunomodulatory effects that were showed in vitro and very small non-randomized studies could be beneficial [25]. Although azithromycin was removed from official guidelines, as well as hydroxychloroquine, shortly after randomized controlled trials demonstrated a lack of any clinical effect, its uncritical use persists in medical practice. However, there is no rational explanation for such azithromycin use in dental medicine. Azithromycin is not the first-line antibiotic in dentistry according to various guidelines and recommendations, and only a few countries use it among the top five antibiotics for odontogenic infections. These include the USA, Brazil, and Belgium [26]. This practice is probably not justified, but the exact causes remain to be determined.

These results on antibiotic utilization are worrying as they demonstrate a continuation of the upward trend in the use of broad-spectrum antibiotics noted by Šutej et al. [12] (for the period 2014–2018), and suggest that restrictive measures may not have been the only reason for this outcome. Because the results of the first study indicated inappropriate antibiotic use in Croatia, the authors initiated changes in the curriculum of the School of Dental Medicine of the University of Zagreb, and in continuing education for dental practitioners a year ago, and have also published therapeutic doses and recommendations for dentists this year. However, it is still too early to see the results of these changes in practice. It is therefore important to continue to monitor dental antibiotic prescribing practices in Croatia, to assess whether or not the increase observed in this study is solely due to the pandemic. If trends continue, more intensive measures and comprehensive efforts at the national level will be needed to maintain efficacy and prevent the possible emergence of antibiotic resistance. 

Croatia is not the only country where dental antibiotic prescriptions increased during the pandemic. Increases were also reported in Scotland (49%) as well as England (12.1%), France (17%), and Canada (76%) [14,17,27,28]. In contrast, a decrease in antibiotic prescribing at the onset of the pandemic was reported in primary dental care in Australia (20%) and antibiotics were prescribed for only 7% of patients in Qatar [29]. Further research is needed to understand the reasons for these differences between countries.

An increase in prescriptions for analgesics was not unexpected. Because orodental problems are frequently accompanied by pain, the use of analgesics in dental medicine is on the rise. They are certainly the first choice for relieving discomfort, inflammation, and pain. These properties were very useful during the pandemic especially for non-urgent odontogenic problems, where anti-inflammatory drugs could reduce the inflammation and buy the time needed to reach the dentist once the freedom of movement had been restored. The trend of increasing prescription of analgesics, especially ibuprofen as the second most prescribed drug in dental medicine in our study, has been observed worldwide for years [16]. The widespread use of analgesics is increasing presumably because of their accessibility and relatively low side effects, and this trend continued and intensified during the pandemic [12,16,30]. While other countries have seen a worrying increase in opioid analgesic prescriptions [13,17,30], our data did show an increase, but the overall consumption of opioids at the national level is low and not a cause for concern. This is likely due to the strict regulations and legislation on opioid prescribing in Croatia. 

Dentists should pay special attention to prescribing practices during the pandemic period, and dental prescribing must remain evidence-based to ensure rationality and good practice. Universal access to high-quality emergency dental care is necessary to avoid non-rational prescribing. The current situation can be seen as an opportunity to develop new tools and approaches. In general, the entire profession can be more oriented toward prevention. Patients should be warned and educated against self-medication with antibiotics, while practitioners should educate patients and raise their awareness of evidence-based dentistry.

It also appears that COVID-19 remains a public health threat, despite widespread control measures implemented to end the pandemic [31]. It may be a good time to incorporate learning outcomes into dental school curricula and continuing education courses for practicing dentists, to include natural disasters and pandemics. This should serve to ensure more rationality and improved clinical practice even in emergencies, with the expectation that this will lead to more rational prescribing of medications.

### Limitations of the Study

A limitation of this study is the lack of information on concomitant diagnosis for prescribed medication. This limitation is due to the separated databases of CIPH and CHIF. Merging the two datasets would provide more specific data on rational drug utilization in the future. In addition, information on private prescriptions is generally missing, and they do not indicate whether the private prescriptions were issued by a GP or a dentist. However, their participation at the national level seems negligible. The last limitation concerns the consumption of OTC (over the counter) medicines because information on their utilization is not recorded in the official database, which would provide better insight into general consumption, especially for analgesic medications. OTC prescriptions at the national level account for 10.5% so the representation of medication consumption even without OTC prescriptions is a realistic consumption rate. OTC medications do not include antibiotics, since their prescribing and dispensing is strictly regulated by law.

## 5. Conclusions

The COVID-19 pandemic has led to a significant change in dental prescribing in Croatia. Restricted access to dental care due to COVID-19 led to increased prescribing of broad-spectrum antibiotics, which reinforces the preexisting upward trend. Within the overall difference in prescribing before and during the pandemic, there is a possibility that antibiotics and pain relief medication prescribing were used as alternatives to active surgical treatment to reduce aerosol generation. Antibiotics are not a substitute for dental procedures and in this aspect, there is a requirement for continued monitoring and evaluation with the goal of reversing the trend through patient and dentist education. Antibiotic prescribing must remain evidence-based to maintain the efficacy of them.

## Figures and Tables

**Figure 1 antibiotics-12-00111-f001:**
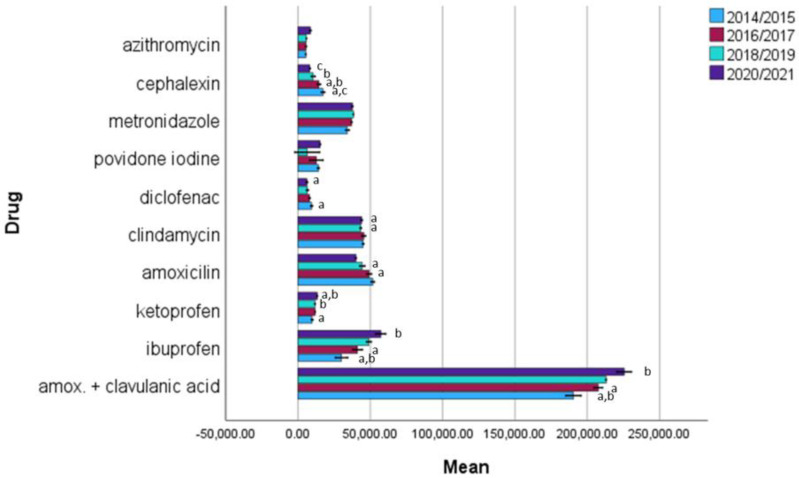
Descriptive statistics of drug prescription in time intervals (2014/15, 16/17, 18/19, 20/21). Same letters indicate statistically significant values for comparisons among year intervals. Error bars denote ±1 SD.

**Table 1 antibiotics-12-00111-t001:** National medication consumption in dental medicine between 2019 and 2021.

Year	Cost */Change Between the Years	Number of Issued Prescriptions (NIS)/Change Between the Years
2019	12,683,256.41	451,728
2020	13,199,189.85/(+4%)	467,947/(+3.5%)
2021	13,507,140.65/(+2.3%)	479,427/(+2.4%)

* Croatian kuna.

**Table 2 antibiotics-12-00111-t002:** The 10 most frequently prescribed medications in DID for period 2014–2021.

Prescribed Medicine	2014 ** DID *	2015 ** DID *	2016 ** DID *	2017 ** DID *	2018 ** DID *	2019 ** DID *	2020 DID *	2021 DID *
amox. + clavulanic acid	1.15	1.22	1.28	1.33	1.36	1.38	1.47	1.54
ibuprofen	0.19	0.23	0.27	0.3	0.33	0.35	0.39	0.43
ketoprofen	0.13	0.14	0.17	0.17	0.17	0.17	0.196	0.2
amoxicilin	0.19	0.19	0.19	0.18	0.174	0.16	0.15	0.16
clindamycin	0.13	0.13	0.13	0.13	0.13	0.128	0.138	0.136
povidone iodine	0.1	0.1	0.11	0.11	0.1	0.09	0.118	0.115
diclofenac	0.15	0.14	0.13	0.12	0.11	0.098	0.1	0.097
metronidazole	0.07	0.08	0.08	0.08	0.08	0.082	0.085	0.081
cephalexin	0.05	0.05	0.05	0.04	0.04	0.031	0.029	0.028
azithromycin	0.01	0.011	0.011	0.012	0.012	0.013	0.016	0.02

* DID daily defined doses per 1000 inhabitants per day. ** This study was based upon a part of the same subject group dataset of a previous publication [12]. The papers substantially add to each other to warrant publication as separate papers.

**Table 3 antibiotics-12-00111-t003:** Most commonly prescribed medications in dental practice by the annual number of prescriptions in 2014–2021.

Prescribed Medicine	2014 ** NIS *	2015 ** NIS *	2016 ** NIS *	2017 ** NIS *	2018 ** NIS *	2019 NIS *	2020 NIS *	2021 NIS *
amox. + clavulanic acid	186,698	194,123	205,411	209,632	212,860	213,156	221,922	229,072
ibuprofen	27,133	33,084	38,757	43,477	47,906	50,166	54,772	59,652
clindamycin	44,805	45,391	46,362	44,544	43,699	43,049	44,333	43,602
amoxicilin	52,492	50,863	50,349	48,077	45,712	43,184	39,796	40,260
metronidazole	33,330	35,049	36,923	37,254	38,284	38,391	37,864	37,308
povidone iodine	13,627	14,471	15,923	9487	294	12,453	15,488	15,224
ketoprofen	9491	10,168	11,834	11,879	11,667	11,883	13,269	13,437
azithromycin	5311	5473	5398	5767	5794	5872	8181	8892
cephalexin	18,092	16,516	15,220	13,796	11,475	9642	8361	8056
diclofenac	9791	8958	8241	7721	6985	6119	6481	5882

* NIS—number of issued prescriptions. ** This study was based upon a part of the same subject group dataset of a previous publication [12]. The papers substantially add to each other to warrant publication as separate papers.

## Data Availability

Data are available by request from the corresponding author.

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
