# Peer review of "Changes in Medication Prescribing Due to COVID-19 in Dental Practice in Croatia—National Study"

_antibiotics, 2023, doi:10.3390/antibiotics12010111_

Round 1
Reviewer 1 Report
You said the pandemic caused significant changes in dentists' activities. Describe more about the new procedures approved in Croatia or what changes took place in the Introduction section.
Replace the comma from Table 2 in case of using it for decimal numbers.
The missing information on diagnosis for prescribed medication is a big limitation because no real conclusions could be taken. The authors just observed an increased medication prescribing, with a low increase in opioid consumption.
The results for Croatia are in the same trend as in other countries during the pandemic and the authors compared them. Specify the name of the countries and what differences were observed.
Author Response
You said the pandemic caused significant changes in dentists' activities. Describe more about the new procedures approved in Croatia or what changes took place in the Introduction section.
We thank reviewer for this suggestion, giving us opportunity to be more clear in introduction story. The part about changes in dentists activities in Croatia has been added in Introduction.
Replace the comma from Table 2 in case of using it for decimal numbers.
Table 2. and Table 3. have been improved with longer string of data.
The missing information on diagnosis for prescribed medication is a big limitation because no real conclusions could be taken. The authors just observed an increased medication prescribing, with a low increase in opioid consumption.
The authors agree with the reviewer on this comment. Missing information on diagnosis is limitation of the study, which authors are still trying to change on national level, since data about indications are collected by National institute for public health, while information about prescriptions issued are collected by Croatian Health Insurance Fund. These two databases are not collaborating and sharing data, and this is a problem that is necessary to solve. One way to solve the problem is to write about it in pharmacoepidemiological research, which we are already doing and also presenting it on national and international conferences. The aim of this study was to observe changes in trend of medication utilization during pandemic period, which showed changes in trends. This result is important for us as educators and clinician so we can be better prepared for the future in rational medication utilization, if unexpected circumstances happen again.
The results for Croatia are in the same trend as in other countries during the pandemic and the authors compared them. Specify the name of the countries and what differences were observed.
We thank reviewer for this suggestion, and we have named countries and their trends.
Reviewer 2 Report
This paper could be interesting, but if the goal is to study how only Covid has affected prescribing, it should be monitored prescription the drugs over a longer period of time. Namely, in this paper the authors only monitored three years, 2019, 2020 and 2021. In my humble opinion, the authors should analysed data from yt least five years (or ten even better). Namely, the increase in prescription of drugs can only be a trend that has nothing to do with Covid.
Conclusion: In my opinion, the authors need to do a more detailed study and observe the trend over a longer period of time.
Author Response
This paper could be interesting, but if the goal is to study how only Covid has affected prescribing, it should be monitored prescription the drugs over a longer period of time. Namely, in this paper the authors only monitored three years, 2019, 2020 and 2021. In my humble opinion, the authors should analysed data from yt least five years (or ten even better). Namely, the increase in prescription of drugs can only be a trend that has nothing to do with Covid.
Conclusion: In my opinion, the authors need to do a more detailed study and observe the trend over a longer period of time.
We thank the reviewer for this useful comment, and we do agree with the suggestion to add data. We have added longer string of data and preformed statistical analasys for the given period, which improved scientific soudness of the manuscript.
Reviewer 3 Report
Abstract: clinical relevance should be provided
Introduction – should address further the importance of the study and the background, as well in Croatia as worldwide
Statistical data should be further analyzed, not just basic percentages
Table 2 and 3 – remove “.” from legend
Discussion: lines 129-135 – concrete examples of the countries and their guidelines should be provided
Lines 137-140 – a detailed comparison should be made to other countries
Lines 199-203 – on what scientific data is the assumption made: there are no references: “as it seems that the pandemic is not going away any time soon”
Clinical relevance for dentists in Croatia and worldwide should be provided
Lines 211-214 should be further developed: consumption of OTC (over the counter) drugs
Conclusion is too vague and should be further concise, based on concrete results
References should be in journal style
Author Response
Abstract: clinical relevance should be provided
We thank reviewer for this usefull suggestion and have added a clinical relevance to the abstract.
Introduction – should address further the importance of the study and the background, as well in Croatia as worldwide
We thank reviewer for this suggestion, giving us opportunity to be more clear in introduction story. The part about changes in dentists activities in Croatia has been added in Introduction.
Statistical data should be further analyzed, not just basic percentages
We thank the reviewer for this useful comment, and we do agree with the suggestion to add data. We have added longer string of data and preformed statistical analasys for the given period, which improved scientific soudness of the manuscript.
Table 2 and 3 – remove “.” from legend
Points have been removed.
Discussion: lines 129-135 – concrete examples of the countries and their guidelines should be provided
Authors tried to gave short insight for other countries, focusing on the aim of this study and country. Since there is overproduction of the publications dealing with the Covid, authors focused on the systematic reviews to get the most informations about the topic.
Lines 137-140 – a detailed comparison should be made to other countries
We thank reviewer for this suggestion, and we have named countries and their trends and made a more detailed comparision.
Lines 199-203 – on what scientific data is the assumption made: there are no references: “as it seems that the pandemic is not going away any time soon”
The references for this assumption have been added and rephrased the statement according to the reference cited.
Clinical relevance for dentists in Croatia and worldwide should be provided
Clinical relevance was added to the manuscript.
Lines 211-214 should be further developed: consumption of OTC (over the counter) drugs
There is no data of OTC medications that are attributed solely to the dentists, and it also represents one of the limitations of the study. Since majority of general prescriptions in Croatia analyzed in this study (89,5%) were funded by the CHIF, presentation of medication consumption according to the CHIF data represents a realistic consumption rate.
Conclusion is too vague and should be further concise, based on concrete results.
The authors agree with the reviewer on this comment and tried to made more concise conclusion based on statistical analysis preformed.
References should be in journal style
The references have been thoroughly checked and adapted.
Round 2
Reviewer 2 Report
Please explain why the new author was added and what was the exact contribution of the new author in the revised manuscript.
There is no changes in methodology, or extensive changes in manuscript writing that the addition of a new author would be justified.
Author Response
We appreciate the opportunity to revise this work and the efforts of the reviewers to improve the clarity of our story.
Since most concerns, criticism and suggestions was for statistical analisys of data, we had consulted a specialist from that field, assistant Luka Šimunović DMD. Since he gave his notable contribution in suggestions and revision of the manuscript, preformed statistical analysis, we included him in author-team.
Reviewer 3 Report
The paper has been improved, according to the reviewer's recommendations!
Author Response
Thanks a lot.